# The Genetic Basis of Non-Contact Soft Tissue Injuries-Are There Practical Applications of Genetic Knowledge?

**DOI:** 10.3390/cells13221828

**Published:** 2024-11-05

**Authors:** Beata Borzemska, Paweł Cięszczyk, Cezary Żekanowski

**Affiliations:** 1Department of Neurogenetics and Functional Genomics, Mossakowski Medical Research Institute, Polish Academy of Sciences, 02-106 Warsaw, Poland; 2Faculty of Physical Education, Gdansk University of Physical Education and Sport, Górskiego 1 Street, 80-336 Gdansk, Poland; pawel.cieszczyk@awf.gda.pl (P.C.); c.zekanowski@imdik.pan.pl (C.Ż.)

**Keywords:** athletes, soft tissue injury, EIMD, ACLR, anterior cruciate ligament rupture, Achilles tendinopathy, Achilles tendon rupture, creatine kinase, exertional rhabdomyolysis, polygenic

## Abstract

Physical activity increases the risk of non-contact injuries, mainly affecting muscles, tendons, and ligaments. Genetic factors are recognized as contributing to susceptibility to different types of soft tissue injuries, making this broad condition a complicated multifactorial entity. Understanding genetic predisposition seems to offer the potential for personalized injury prevention and improved recovery strategies. The candidate gene analysis approach used so far, has often yielded inconclusive results. This manuscript reviews the most commonly studied genetic variants in genes involved in the musculoskeletal system’s structure and recovery processes (*ACTN3*, *ACE*, *CKM*, *MLCK*, *AMPD1*, *IGF2*, *IL6*, *TNFα*, *CCL2*, *COL1A1*, *COL5A1*, *MMP3*, and *TNC*). Referring to the literature, it was highlighted that single-gene analyses provide limited insight. On the other hand, novel genetic testing methods identify numerous variants of uncertain physiological relevance. Distinguishing between functionally important variants, modifying variants, and the thousands of irrelevant variants requires advanced bioinformatics methods and basic multiomics research to identify the key biological pathways contributing to injury susceptibility. Tools like the Total Genotype Score (TGS) and Polygenic Risk Score (PRS) offer a more holistic view by assessing the combined effect of multiple variants. However, these methods, while useful in research, lack clinical applicability. In conclusion, it is too early to determine the clinical implications of genetic variability as a tool for improving well-established training and injury prevention methods, as the predictive power of genetic testing for injury predisposition is currently low.

## 1. Introduction

Physical activity, whether amateur or professional, is associated with an increased risk of injury. Most injuries sustained are non-contact (66.6%) and involve muscles, tendons, and ligaments [1]. Muscle injuries account for 20–30% of all time-loss injuries at the professional level and up to 23% at the amateur level. Approximately 16% of muscle injuries are recurrent [2]. The lifetime prevalence of tendon injuries is up to 23.9% among athletes and 5.9% in the general population. Ligament injuries also affect millions of individuals [3]. A familial predisposition to anterior cruciate ligament (ACL) injury has been uncovered, suggesting that having a relative with an ACL rupture (ACLR) doubles the risk of the same ligament injury [4]. A similar conclusion can be drawn from twin studies [5]. Thus, ACL injuries are estimated to have a significant hereditary component, with heritability of ca. 69% [6]. Furthermore, it is observed that, on average, 65% of patients resume their previous level of sports participation, and only 55% return to competitive-level sports following surgical intervention [6].

In light of the high prevalence of muscle, tendon, and ligament injuries and the substantial healthcare costs and associated clinical consequences, it is imperative to gain insight into the underlying mechanisms. A more comprehensive grasp of the risk factors and etiology may benefit future risk screening.

The treatment and prevention of sport injuries is a high priority for athletes, coaches, trainers, and medical personnel [7,8]. Due to the multifactorial nature of injuries, high-performance personnel develop training and rehabilitation programs tailored to specific actions and movements, taking into account the combination of physical characteristics and the technical skill of the athlete. Given that athletes with prior injuries are susceptible to recurrence, decisions regarding their return to performance can have significant consequences for both the players and their teams [9,10]. Physical resilience is a complex phenotype influenced by both biological and environmental factors. Among the external factors, training and nutrition should be emphasized. On the other hand, individual factors such as age, sex, body structure, anatomical variability, mobility, biomechanics, neuromuscular control, previous trauma, and fitness level determine susceptibility to soft tissue injury [11,12]. A large proportion of inter-individual variability is affected by genetic background.

According to Baumert et al. (2019), the ability to succeed at the highest level of sport may depend on the susceptibility to non-contact injury of muscle fibers, the extracellular matrix (ECM) surrounding the entire skeletal muscle, the resilience of tendons and ligaments, and regenerative capacity [13,14]. Consequently, significant research has been conducted in the area of genetic variation, focusing on the aforementioned processes over the past two decades [15,16]. The possibility of creating individualized training programs based on an athlete’s genetic predisposition to injury appeared to be a breakthrough in current training methods [7,8]. The vast majority of proposed risk factors for musculoskeletal injuries have been generated using the hypothesis-driven candidate gene approach, in which a case–control cohort is screened for variants in genes of interest [8]. Lack of replication or even ambiguous results may be explained by the polygenic nature of the trait or by the methodology focusing only on common variants. It has been suggested that the hypothesis-free approach, based on GWAS or whole-genome sequencing, may reveal new mechanisms underlying function and pathology in muscles, tendons, and ligaments.

The main goal of the research is to identify and understand the genetic and resulting molecular and cellular mechanisms that may have clinical applications. To date, twenty-nine genetic markers have been associated with soft-tissue injuries in at least two studies [17]. This manuscript reviews the most commonly studied genetic variants that are thought to predispose athletes to a higher/lower risk of exercise-induced muscle damage, as well as susceptibility to tendon and ligament injuries. We have compiled a detailed analysis of the study groups and methodology used in the literature to consider the validity of using available genetic knowledge in personalized injury prevention and recovery. In this review, we focused on the most promising and extensively analyzed genetic markers: *ACTN3* rs1815739, *ACE* rs4341, *CKM* rs8111989, *MLCKI* rs2700352, *MLCKII* rs28497577, *AMPD1* rs17602729, *IGF2* rs3213221, *IGF2* rs680, *IL6* rs1800795, *TNFα* rs1800629, *CCL2* rs2857656, *COL1A1* rs1800012, *COL5A1* rs12722, *MMP3* rs679620, and *TNC* rs2104772. Sports specialists are interested not only in the efficiency of the musculoskeletal system but also in the predisposition to increased risk of injury and the ability to recover. The genes with the strongest scientific support encode both structural and functional proteins of muscle, the extracellular matrix, ligaments, and tendons, as well as proteins involved in post-exercise recovery pathways [18]. Variations in these genes are thought to explain the great variability in exercise-induced soft tissue damage.

## 2. Exercise-Induced Muscle Damage

The mechanism of exercise-induced muscle tissue damage (EIMD) is well understood. Overexertion caused by eccentric contractions can result in disruption of the sarcomere and the extracellular matrix, resulting in inflammation. EIMD is characterized by a reduced ability to generate force and a reduced range of motion, tenderness that develops into delayed-onset muscle soreness (DOMS) within 24–72 h, and increases in intramuscular proteins in systemic circulation, including creatine kinase and myoglobin. Overexertion and insufficient muscle recovery, especially in hot environments, can lead to massive muscle tissue breakdown and fatal response due to myoglobin-induced renal failure and/or cardiac arrhythmia [19]. There is a high degree of variability in the damage response of individuals, with higher responders at risk of potentially fatal rhabdomyolysis [20]. Besides such extreme cases, individuals also differ in muscle stiffness or the severity of DOMS in response to eccentric contractions. In addition to exercise adaptations resulting from training level, the importance of genetic variability is emphasized.

## 3. Variants Associated with Muscle Structure and Function

### 3.1. rs1815739 ACTN3

The most studied variant in terms of sporting success [21] and injury susceptibility is rs1815739 (R577X) in the alpha-actinin-3 (*ACTN3*) gene. ACTN3 is one of the main structural components of the contractile apparatus at the Z-disk in type II (fast) muscle fibers. It is responsible for crosslinking and anchoring actin filaments [22,23]. About 18% of Western Europeans lack this protein due to homozygosity for a premature stop codon 577XX [24]. This can affect the structural integrity of sarcomeres and muscle cells during muscle contractions, as Z-disks are particularly susceptible to damage.

There is an 80% similarity between α-actinin 2 and α-actinin 3, suggesting that ACTN2 could substitute for ACTN3 [25]. However, in human muscle, ACTN2 is expressed in all muscle fibers, while ACTN3 is only found in fast, glycolytic muscle fibers, which are responsible for rapid force generation. ACTN3 may provide type II fibers with a greater ability to absorb and transmit force at the Z-line during fast contractions [24]. The absence of *ACTN3* results, e.g., in lower muscle strength, suggesting that ACTN2 is not fully compensating at the extremes of performance [26].

The observed changes in the interaction of Z-disk proteins in *ACTN3* knockout muscles, that alter their elastic properties, may explain why individuals with the 577XX genotype are more susceptible to contraction-induced muscle damage [23]. Moreover, the XX homozygotes present greater signs of exertional rhabdomyolysis than their RR counterparts [19]. Following eccentric training, athletes with the XX genotype exhibited elevated serum concentrations of creatine kinase, α-actin, and myoglobin, a heightened degree of postexercise repair response, elevated levels of self-perceived local muscle pain, and augmented muscle fatigue as signs of muscle damage [26,27]. A systematic review of 13 studies involving 1093 participants indicated that the *ACTN3* XX genotype may predispose athletes to a higher probability of some non-contact injuries and exercise-induced muscle damage [28]. Furthermore, Clos et al. (2019) observed a tendency towards a longer recovery period following moderate muscle injury in players with the 577XX genotype compared to those with the 577RR and 577RX genotypes (i.e., 41, 22, and 33 days, respectively). Rodas et al. (2021) reported that players with 577RR and 577RX genotypes returned to normal competition after an average of 17–20 days, while those with the XX genotype required an average of 36 days to do so [29,30].

A deficiency in ACTN3 has been linked to higher body fatness, lower fat-free mass volume, lower muscle strength, and higher muscle flexibility and range of motion [31]. Moreno et al. (2020) suggest that a deficiency in ACTN3 may result in both positive and negative phenotypes with regard to endurance running. This would explain why a loss-of-function genetic variant has been positively selected through evolution [32]. *Actn3* knockout mice exhibit a reduction in fast type II muscle fiber size, a shift towards a slow-twitch aerobic metabolic phenotype, and increased glycogen storage and mitochondrial oxidative enzyme activity [33]. Moreno et al. (2020) observed that 577RR marathoners were 1.88 times more likely to experience an injury related to endurance training or competition. In contrast, 577XX marathoners exhibited a higher frequency of sudden injuries and were 2.86 times more likely to sustain a muscle-type injury [32]. The influence of the 577X allele on exercise-induced muscle damage remains inconclusive. While 577XX homozygosity has been consistently linked to adverse outcomes in the majority of studies, there are emerging reports suggesting a neutral or even beneficial influence.

### 3.2. rs8111989 CK-MM

In the research, the most widely used marker to evaluate the level of exertional rhabdomyolysis is the serum concentration of the enzyme creatine kinase [19]. Muscle-specific creatine kinase (CK-MM) plays an essential role in maintaining muscle energy homeostasis by rephosphorylating ATP, thereby influencing skeletal muscle’s capacity to resist strenuous exercise. CK-MM is localized at the M-line in myofibrils. It has been shown that type II (fast twitch) fibers exhibit significantly higher CK-MM activity compared to type I (slow twitch) fibers [34].

The rs8111989 variant located in the 3′-untranslated region of the *CK-MM* gene [35], could affect the rate of mRNA translation. The rs8111989 G variant has been demonstrated to reduce the activity of the skeletal muscle in endurance athletes, putatively leading to energy depletion and metabolic insult [20,36]. However, the results of analyses conducted on athletes are inconclusive. Specifically, the rs8111989 AA homozygotes had a six-fold higher risk of being a high CK responder after a stepping exercise [37], whereas another study found that GG homozygotes were 3.1 times more likely to experience exertional rhabdomyolysis than carriers of the A variant [20].

### 3.3. rs17602729 AMPD1

The *AMPD1* gene encodes adenosine monophosphate deaminase, which is a crucial enzyme involved in energy production in skeletal muscles, particularly during intense physical exercise. AMPD1 catalyzes the deamination of adenosine monophosphate (AMP) into inosine monophosphate (IMP). This process reduces the accumulation of adenosine diphosphate (ADP) and shifts the reaction toward adenosine triphosphate (ATP) production [38,39].

A common C-to-T transition at position 34 of *AMPD1*, rs17602729, has been observed in European Caucasians at a rate of 11%. This variant results in the generation of a premature stop codon. A deficiency of the AMPD1 enzyme results in exercise-induced myopathy. Individuals carrying the rs17602729 TT genotype are susceptible to a number of adverse effects, including muscle cramps, increased perceived pain following training, delayed recovery of muscle strength, and early fatigue during exercise [36,39]. Collins et al. (2017, 2018) observed that snooker players with the CC genotype of the *AMPD1* gene exhibited significantly reduced incidents of injuries [40]. Conversely, weightlifters with the rs17602729 T variant experienced more pain following training and required extended periods of recovery [39]. The detrimental effects of the rs17602729T variant have been confirmed in other studies.

### 3.4. rs28497577 and rs2700352 MLCK

Myosin light chain kinase (MLCK) is involved in muscle force generation, as MLCK phosphorylates the light chain of myosin, particularly in Type II fibers [20]. This leads to increased force generation during skeletal muscle contraction. It has been postulated that the rs28497577 C and rs2700352 T variants may influence light chain phosphorylation, which could putatively reduce the capacity to generate force and withstand strain during voluntary muscle contractions. Therefore, these genetic variations may predispose individuals to muscle damage during endurance competitions [36,41]. Childers and McDonald (2004) demonstrated that the phosphorylation of the light chain increases the injury caused by eccentric contractions in skinned type II fibers of the rat psoas muscle. Additionally, they considered the possibility that MLCK influences susceptibility to contraction-induced damage by affecting the structural integrity of the myofibrillar lattice [42].

### 3.5. rs4341 ACE

The second most investigated exercise-related genetic variant is the functional insertion/deletion (I/D) polymorphism in the angiotensin-converting enzyme (*ACE*) gene [43]. In the existing literature, the D variant is thought to provide an advantage in injury prevention. ACE plays a pivotal role in the renin-angiotensin system (RAS) and the tissue kallikrein-kinin system (TKKS). In the renin-angiotensin system, ACE catalyzes the conversion of angiotensin I into angiotensin II (ANG II) [44]. ANG II exerts a multitude of effects, including stimulation of vascular smooth muscle growth, capillary density, oxygen consumption, modulation of neuromuscular transmission, and hypertrophic effects on skeletal muscle, thereby enhancing contractile function. In the tissue kallikrein-kinin system, ACE catalyzes the degradation of bradykinin to biologically inactive fragments. Bradykinin lowers blood pressure by vasodilation and induces local inflammation and the extravasation of myocellular proteins. The RAS and KKS systems play an important role in the inflammation and muscle damage induced by exercise [45].

ACE I/D polymorphism has also been linked to a variety of exercise-related phenotypes, including muscle strength, metabolism, volume, cardiac growth response to exercise, differences in skeletal muscle fiber distribution, composition, and capillarization [44]. Furthermore, elevated ACE activity associated with the DD genotype has been shown to enhance blood supply, thereby facilitating the remodeling of damaged muscle structures and improving muscle recovery [13,46]. Elevated serum CK activity in carriers of the II genotype indicates an impaired regenerative process and may also increase the risk of developing exertional rhabdomyolysis [37,45,47]. It has been proposed that the II genotype increases the risk of developing muscle damage, whereas the DD genotype may confer protective effects against exercise-induced muscle injury. However, many studies have failed to show a significant effect of *ACE* gene variability on injury susceptibility. In a recent study, de Almeida and colleagues found that the ACTN3 577X variant and ACE II genotype collectively result in a 100% increase in the incidences of injuries per season (1.682 injuries/season versus 0.868 injuries/season) [48].

## 4. Inflamation and Regeneration-Related Variants

### 4.1. rs2857656 CCL2

Regeneration is the process of recovery and growth of muscle tissue after exercise. Ineffective regeneration increases the risk of further injury or chronic post-injury conditions. Chemokine ligand 2 (CCL2), known to be produced by both macrophages and satellite cells, plays a key role in inflammation and immunoregulation [49]. CCL2 is involved in the attraction of monocytes, memory T cells, dendritic cells, macrophages, and fibroblasts to the site of injury and the subsequent upregulation of collagen type I expression [10,15]. CCL2 expression increases dramatically after muscle injury and is involved in muscle repair and the adaptive response to chronic exertion [49]. Mice lacking the CCL2 receptor have been shown to have impaired regeneration following freeze injury [50]. As a result of these processes, repetitive strength training may improve the integrity of the muscle extracellular matrix (ECM) over time, enhancing strength transmission but also stiffness. A stiffer ECM may be more susceptible to damage during high-speed eccentric contractions [15]. The rs2857656 GG genotype has been suggested to be associated with a higher baseline muscle strength [51], but also with more severe muscle injuries [52].

### 4.2. rs1800795 IL-6

Cytokines play an important role in inflammation following EIMD, acting as mediators of communication between immune cells and regulating the influx of inflammatory cells to the site of injury [53]. It has been reported that prolonged and/or intense exercise can significantly increase circulating levels of cytokines such as interleukins [54]. Interleukin-6 (IL-6) plays a key role in systemic inflammation by mediating other cytokines. However, IL-6 has a complex role and is associated with both pro- and anti-inflammatory responses. IL-6 is also strongly associated with glucose transport in human skeletal muscle, indicating a sustained need for glucose to repair damaged muscle fibers and muscle ECM [55].

The rs1800795 G variant located in the promoter region within the *IL-6* gene seems to increase *IL-6* gene transcription [56]. Larruskain et al. (2018) found that the rs1800795 GG genotype was associated with a higher risk of hamstring injury [57], while the rs1800795 CC genotype was associated with higher CK levels after eccentric exercise in healthy individuals [58]. However, most studies did not find an association.

### 4.3. rs1800629 TNF-α

There is increasing evidence that IL-6 has anti-inflammatory effects, in part by inhibiting the production of the pro-inflammatory cytokine, tumor necrosis factor (TNF-α). TNF-α is expressed primarily in lymphocytes and plays a crucial role in the remodeling of muscle structure in the context of EIMD by initiating the breakdown and subsequent removal of damaged muscle fragments [59]. A single nucleotide substitution, rs1800629 G/A in the promoter region, regulates *TNF* gene expression. Macrophages of the rs1800629 GG genotype are thought to potentially secrete more local TNF after infiltrating the injury site hours after EIMD [58].

### 4.4. rs3213221 and rs680 IGF2

An important mediator of skeletal muscle adaptation is insulin-like growth factor 2 (IGF2), which functions in an autocrine/paracrine mode. Keller et al. (1999) found that after contact-induced injury in mouse soleus muscle, there was a significant increase in IGFs in the days following the injury [60]. IGF2 is produced in several tissues, including skeletal muscle and active immune cells, and is involved in satellite cell differentiation, proliferation [61], and ECM integrity [60]. The majority of these SNPs are reported to be associated with immediate strength loss, pain, and serum CK response. Chronically elevated IGF2 expression may lead to malformation of muscle tissue structure [62,63]. rs3213221 and rs680 variants are mostly analyzed in the context of sports injuries.

## 5. Variants Associated with Extracellular Matrix Components

### 5.1. rs679620 MMP3

Extracellular matrix metalloproteinases (MMPs) are relevant to the functionality of the musculoskeletal system. The MMPs comprise zinc-dependent endopeptidases. These enzymes play a crucial role in the remodeling of the extracellular matrix, which affects skeletal muscle recovery by degrading ECM proteins [64]. The balance between collagen deposition and degradation is critical for scar tissue formation and muscle stiffness [15].

Variations in the expression of *MMPs* are thought to alter the function of the ECM and influence muscle regeneration [65,66]. In this regard, the rs679620 variant of the *MMP3* has received the most attention from researchers. The functional rs679620 variant is a non-synonymous SNP (Glu45Lys) in exon 2, although this polymorphism has not been predicted to be damaging. The rs679620 C variant is thought to decrease *MMP3* transcription. The *MMP3* rs679620 T-variant is in linkage disequilibrium with the *MMP3* rs3025058 5A functional variant within the promoter region [1]. The 5A allele is reported to have increased transcriptional activity [67].

Conclusions from the studies of the rs679620 variant in relation to sports injuries are conflicting. Raleigh et al. (2009) demonstrated the association of the rs679620 CC genotype with Achilles tendinopathy but not with Achilles tendon rupture in South African Caucasians [68]. However, in the British cohort, this variant was associated with Achilles tendon rupture [69]. More recently, the rs679620 C allele was associated with an increased risk of anterior cruciate ligament rupture [70]. Interestingly, the opposite rs679620 TT genotype was associated with hamstring injuries [57]. Previously, the rs679620 TT genotype was associated with a risk of chronic Achilles tendinopathy in two independent study cohorts from Australia and South Africa [71], and with anterior cruciate ligament ruptures in South African Caucasians [72]. Although the rs679620 variant showed a significant association with soft tissue injuries, there were contrasts in the direction of association reported by the studies. This can suggest that the variant acts as a proxy for the true risk-conferring variant to which it is genetically connected.

### 5.2. rs2104772 TNC

Extracellular matrix (ECM) glycoproteins play a pivotal role in the development, structure, and function of both tendons and ligaments. Tenascin-C (TNC) is one of the glycoproteins crucial to the properties of the extracellular matrix. It plays an important role in tissue remodeling, regulating cell–matrix interactions and the damage-repair cycle, and providing strength and elasticity in contact with mechanical forces. TNC is strongly induced by infection and inflammation. Accordingly, the *tnc* knockout mouse has a reduced inflammatory response. TNC has the potential to modify cell adhesion either directly or through interaction with fibronectin. Cell-tenascin interactions typically lead to increased cell motility [73]. It is involved in angiogenesis and wound healing. TNC is expressed in regenerating myofibers in response to mechanical loading in the myotendinous junction [74]. TNC plays a role in the adaptation of tendon tissue during the initial reactive and tendon degenerative phases of tendinopathy. Subtle alterations in its properties may result in an inappropriate healing response during the early stages of reactive tendinopathy and predispose the tendon to early progression toward advanced stages of tendinopathy [75,76]. The large number of sports-related muscle injuries occurring near the muscle-tendon junction prompted researchers to analyze variability in the *TNC* gene.

The rs2104772 variant has been suggested to contribute to structural instability in the fibronectin type III-D domain of TNC, altering the molecular elasticity of the domain [57,77]. The rs2104772 A variant has been associated with an increased risk for Achilles tendinopathy [76] and hamstring injury [57], and contributes to a haplotype associated with rotator cuff repair [78]. In a Genome-Wide Association Study (GWAS) conducted with a large cohort of the general American population, no genetic association was found when *TNC* variants were examined separately or in combination in the context of Achilles tendon injury and ACL injury [79].

## 6. Collagen Genes Variants

### 6.1. rs12722 COL5A1

Type I and III collagens are the main types that form fibrils in tendons, ligaments, and connective tissue in skeletal muscle. However, type V collagen plays an important role as a structural component of fibrils [80]. It intercalates with type I collagen to form heterotypic fibrils. Type V collagen makes up ca. 10% of the collagen in ligaments. It is a complementary fibrillar collagen involved in the organization and regulation of the diameter of type I collagen fibrils and thus the structural and functional properties of tendons and ligaments [81].

The major isoform of type V collagen is a heterotrimer consisting of two α1 chains and one α2 chain, which are encoded by the *COL5A1* and *COL5A2* genes, respectively. Several mutations within these genes have been shown to cause more severe connective tissue disorders such as Ehlers–Danlos syndrome [82], characterized by joint laxity and connective tissue fragility, emphasizing the significance of this collagen in tissue structure and function. The rs12722 variant in *COL5A1* has been extensively studied with regard to injury risk. The rs12722 T variant is associated with increased mRNA stability by increasing the abundance of collagen type V, which was associated with smaller fibril diameter, increased tissue-fibril density, and tissue stiffness. Therefore, the rs12722 T variant may contribute to a stiffer tendon/muscle ECM, leading to better force transmission, i.e., improved running performance [83]. However, this muscle/tendon ECM might be more susceptible to being overstretched by extreme external forces, making T variant carriers more vulnerable to EIMD and soft tissue injuries compared to the C variant carriers. The rs12722 T variant is a strong risk factor for tendinopathies [68,84,85], Achilles tendon rupture [68], anterior cruciate ligament rupture in females (Michael Posthumus et al., 2009a), tennis elbow [86], and skeletal muscle injuries [10,52,87].

### 6.2. rs1800012 COL1A1

Type V collagen and tenascin C are quantitatively minor tendon components, whereas type I collagen is the major structural component. The major isoform of type I collagen is a heterotrimer consisting of two α1 chains and one α2 chain encoded by the *COL1A1* and *COL1A2* genes, respectively. A functional variant, rs1800012 G/T, is located within the first intron of *COL1A1* in a Sp1 transcription factor binding site. The rare rs1800012 TT genotype is underrepresented in Swedish patients with cruciate ligament ruptures and shoulder dislocations [88]. Other studies in Caucasian South African patients [89] and Polish professional soccer players [90] suggested that the TT genotype may protect against ACL ruptures, contrary to studies in elite Australian football players and Polish skiers, which suggested protection of the rs1800012 GG genotype [10,91].

The detailed literature data for all the variants mentioned above is summarized in Table 1.

## 7. Challenges of Genetic Testing

Candidate gene association studies have often generated inconclusive results (Table 1). The ambiguity in the results of studies conducted over the years by scientific teams may be due to obvious reasons such as ethnic origin, heterogeneity of the population, and the strength of the research determined by the size of the study groups. It is known that sex can predispose individuals to a distinct type of injury. As reported by Larruskain et al. (2018), the total number of absence days was 21% higher in women, with a 5.36 times higher incidence of severe knee and ankle ligament injuries. In men, the incidences of hamstring strains and pubalgia cases were 1.93 and 11.10 times greater, respectively [57]. The incidence of quadriceps strains, anterior cruciate ligament ruptures, and ankle syndesmosis injuries were 2.25, 4.59, and 5.36 times higher in women, respectively [122]. The prevalence of muscle stiffness, which is a risk factor for muscle injury, is lower in females than in males, because estrogen level is inversely correlated with muscle stiffness [123]. In addition, estrogen exerts anti-inflammatory and antioxidant effects on skeletal muscles. It is noteworthy that in a study by Yang et al. (2003), the *ACTN3* XX homozygote was found in 8% of male power athletes, whereas no women carried this null genotype. In contrast, the control sample of women in that report exhibited a 20% frequency of the XX homozygote. This suggests that there may be a selection against the XX homozygous condition in this type of athletic performance, which aligns with data indicating that XX homozygous women had the lowest maximum voluntary contraction (MVC) strength [21]. The findings of Hall et al. (2021) indicate that tissue composition and the proportion of genetic factors contributing to musculoskeletal injury risk change with age [1]. Also, training methods and injury prevention programs may condition injury risk and be a confounding factor for injury studies. Intense damage or incomplete recovery from EIMD are the factors that can increase injury risk [15].

It is worth noting that previous studies have examined the genetic influences on susceptibility to muscle damage in response to a standardized exercise protocol, which is not representative of the physiological conditions encountered in most real-world sporting activities, by measuring changes in creatine kinase activity [37,47,49,58,105,106]. More recent research evaluates real musculoskeletal injury in actual sports situations, considering the work of the entire musculoskeletal system, dehydration, and heat stress. Creatine kinase level in the blood is widely used as a marker to reflect muscle breakdown. However, the relationship between high CK values and clinical symptoms remains unclear [20,124,125,126]. Several studies have observed that prolonged strength loss in the days following eccentric exercise is not associated with an increase in blood CK activities. The initial injury is attributed to mechanical disruption of the fiber, but the subsequent damage is related to inflammatory processes and alterations in excitation-contraction coupling within the muscle [53]. It is well-documented that ultra-endurance practice results in pronounced muscle damage, putatively as a result of mechanical disruption of the fiber, disturbances in calcium homeostasis, and inflammatory processes [98].

Most importantly, the components of the musculoskeletal system are interdependent, which can lead to mistakes in association studies analyzing musculoskeletal injuries as a simple phenotype. For example, an increase in tendon stiffness can result in greater fascicular lengthening of muscles and subsequent muscle damage following eccentric exercise. Conversely, passive muscle stiffness is considered a significant factor affecting joint flexibility [97]. Thus, for example, the hypothesis that ACTN3-deficient athletes may have a higher risk of ligament damage can be attributed to a dysfunction in the muscle’s ability to support the joints during sport-specific movements, rather than any direct impact of the XX genotype on ligament characteristics [28].

## 8. The Polygenic Nature of Musculoskeletal Pathology

As mentioned above, musculoskeletal soft tissue injuries are complex phenotypes with a polygenic contribution that most likely represents a combination of biologically relevant pathways [107]. Elucidating the extent to which sport-related injuries are the result of lifestyle and exposure to factors that cause musculoskeletal pathologies, and the extent to which they are the result of hereditary and genetic conditions of varying intensity, is one of the most important tasks of basic sports science. It should be noted that the hereditary component is not equivalent to simple genetic factors, but also includes DNA methylation and functional RNAs, which play an important role in development and response to environmental stressors. Above, we have reviewed the genetic factors that influence or potentially influence the development of muscle, tendon, and ligament injuries and diseases. While the search for genomic correlates of musculoskeletal pathology is scientifically justified, using it as a way of pinpointing putative and novel molecular pathways and physiological processes whose damage or disfunction may cause the aforementioned class of conditions, genetic diagnostics, and especially commercial (direct-to-consumer) diagnostics, is not a valid approach. 

Moreover, the identification of a single genetic variant provides only a very limited insight into the genetic basis of susceptibility to sport-related injuries, especially in the absence of an understanding of the broader genomic and epigenetic context. The single genome comprises an average of 10 million polymorphic sites, including single nucleotide variations (SNPs), indels, and larger structural variants (SVs). Consequently, reducing the genetic context to the result of a single variant test appears to be an implausible approach [127]. The utility of genetic tests that identify specific, single-gene variants was analyzed in detail more than two decades ago [128]. For example, concerning the positive predictive value (PPV), i.e., the probability that a particular phenotype, such as musculoskeletal pathology, will develop in an individual with a positive test result, only rare polymorphisms (with a frequency of about 1% in the general population) that significantly increase the risk of the condition are worth detecting in the general population (PPV > 50%). The second factor to consider when deciding whether genetic testing is appropriate is the proportion of disease cases that can be attributed to a particular genotype, rather than to environmental influences or genes other than the one being tested (the attributional risk, AR). Attributional risk is low for common diseases, because other factors may be critical in the development of a disease. In the case of genotypes that occur at a frequency of 10–30% in the population and increase the risk of disease by a factor of 10–20, the attributional risk is clinically sufficient. Conversely, if a subject has genotypes that confer disease susceptibility in more than one gene, the AR risk associated with a particular gene is much lower.

### 8.1. Total Genotype Score

Furthermore, it is assumed that the genetic basis of phenotypic traits is formed by numerous variants, each of which has a small effect on the phenotype. Considering the low predictive power of individual SNP, the SNPs associated with injury prevalence and/or absence have been incorporated into multifactorial models to determine the risk of acute soft tissue rupture. One approach is to increase the number of variants analyzed and calculate the total genotype score (TGS), which is a measure of the additive statistical relationship between multiple variants and the phenotypic trait of interest. Using this model 2, 1, and 0 points were assigned for optimal, intermediate, and suboptimal genotypes, which were awarded and then accumulated and related to a maximum score of 100 arbitrary units (a.u.) for the most favorable gene set. This approach is most commonly used in soft tissue injury risk analysis. The multifactorial analyses included the most significant variants, which are also discussed in this review.

Massida et al. (2024) included *ACE* I/D, *ACTN3* rs1815739, *COL5A1* rs2722, and *MCT1* rs1049434 to discriminate non-injured from injured football players. The mean TGS of non-injured football players (63.7 ± 13.0 a.u.) was different from that of injured football players (42.5 ± 12.5 a.u., *p* < 0.001). There was a TGS cut-off point (56.2 a.u.) to discriminate non-injured from injured football players. Players with a TGS under this cut-off had an odds ratio of 3.5 (95% CI 1.8–6.8; *p* < 0.001) to suffer an injury when compared to players with a TGS above this value. The *COL5A1* rs2722 variant was also included in the analysis, although it did not show a statistically significant difference between the groups analyzed [87].

Varillas-Delgado et al. (2023) discriminated non-injured athletes (68.263 ± 13.197 a.u.) from injured athletes (50.037 ± 17.293 a.u., *p* < 0.001), and determined a TGS cut-off point (59.085 a.u.) with an odds ratio of 7.400 (95% CI 2.54821.495, *p* < 0.001). *ACE* I/D, *ACTN3* rs1815739, *CKM* rs8111989, *AMPD1* rs17602729, *MLCK* rs2849757, and *MLCK* rs2700352 were included in the multivariate analysis. In the analysis of individual variants, only *AMPD1* rs17602729 showed statistical significance [36].

The same set of variants (*ACE* I/D, *ACTN3* rs1815739, *CKM* rs8111989, *AMPD1* rs17602729, *MLCK* rs2849757, and *MLCK* rs2700352) was used by Maestro et al. (2022) to discriminate TGS in non-injured soccer players (57.18 ± 14.43 a.u.) from that of injured soccer players (51.71 ± 12.82 a.u., *p* = 0.034). Players with a TGS under the cut-off point (45.83 a.u.) had an odds ratio of 1.91 (95% CI: 1.14–2.91; *p* = 0.022) to suffer an injury. Only *AMPD1* rs17602729 and *MLCK* rs28497577 showed statistical significance in the individual analysis [92].

Del Coso et al. (2020) used *ACE* I/D, *ACTN3* rs1815739, *CKMM* rs1803285, *MLCK* rs28497577, *TNFα* rs1800629, *IGF2* rs3213221, and *IL6* rs1800795 variants to show that TGS was higher in low CK responders than in high CK responders, as determined after a half-ironman race (7.7 vs. 5.5; *p* = 0.01; The authors did not relate the maximum score to 100 a.u.). When the individual scores of each gene were considered, only the *ACTN3* R577X variant was statistically higher in low-CK responders. Interestingly, when the genotype score was calculated using only three variants (*ACTN3* rs1815739, *CKMM* rs1803285, and *MLCK* rs28497577) directly related to skeletal muscle fiber properties, low CK responders also had a significantly higher score than high CK responders (2.3 vs. 1.4 points). The remaining four variants may affect ER through inflammatory processes and muscle regeneration. The TGS of these 4 variants also differs between low and high CK responders (5.4 vs. 4.3 points) [93].

In a previous study, Del Coso et al. (2017) used the same genetic variants to calculate TGS, based on post-race CK levels in marathon runners. Low CK responders had a higher TGS than high CK responders (5.2 ± 1.4 vs. 4.4 ± 1.7 points). The differences in the distribution of TGS were noted, although there were no significant differences between low and high CK responders for the individual scores of each variant [19].

It is worth noticing that Hall et al. (2021) included only variants individually associated with injury prevalence (*ACTN3* rs1815739, *MLCK* rs28497577, *IL6* rs1800795, *COL5A1* rs12722, *EMILIN1* rs2289360, *MMP3* rs28497577, and *VEGFA* rs2010963) in very young soccer players. Even though each variant individually showed statistical significance, the difference in TGS between the groups was barely statistically noticeable (*p* = 0.048) [1]. Contrary to the previous studies, players with one or more injuries had a higher TGS than non-injured players (46.5 ± 13.1 vs. 43.9 ± 12.6), because, in this study, a suboptimal genotype receives 2 points.

Indeed, the variance score, as well as the TGS, can distinguish a group of athletes from non-athletes or characterize athletes with different susceptibility to sport-related pathologies, but does not predict the predisposition of any particular athlete [129].

### 8.2. Polygenic Risk Score

Another method for assessing the influence of genetic background on phenotypic traits is the polygenic risk score (PRS), which is a weighted sum of alleles at single-nucleotide polymorphisms that have been identified as significant. The selection of variants and their impact values are defined according to the results of multiple GWAS analyses. This approach allows for the generation of an additive model of the impact of variants on a particular trait of an organism. This model serves as the basis for estimating the impact of variants identified in the test subject.

Genome-wide association screens for Achilles tendon and ACL injury conducted by Kim et al. (2017) did not identify any significant genetic variants contributing to pathology, including those selected in case–control studies [79]. Due to the nature of GWAS analyses, the impact of variants with minimal, additive effects on phenotype may be masked by statistical noise.

The objections raised to simple tests that analyze single variants and TGS also apply to analyses using the PRS, which is an estimate of relative (statistical) risk, not absolute disease risk. It has been noted [130] that the theoretical basis of PRS calculations is an additive model that does not take into account the complex molecular and physiological processes, dependent on environmental factors, that co-determine the formation of phenotypic traits. The American College of Medical Genetics and Genomics (ACMG) position statement of 2023 indicates that PRS results are not a basis for clinical diagnosis, but rather an estimate of clinical risk. Furthermore, a low PRS does not exclude a high risk of a disease or a normal trait [131]. The ACMG states that the meaning of the result should be discussed with the person tested and that the PRS test itself cannot be part of “evidence-based management” (EBM), due to the limited data supporting the method in clinical applications [130]. The regulation of PRS applications within the health service will require the development of novel approaches and the implementation of new risk analyses for potential users [132].

## 9. Conclusions

The molecular and physiological mechanisms underlying different injuries are known to be similar, but the assessment of the degree of similarity still needs to be refined. Our review suggests that the genetic signature may differ between different types of musculoskeletal soft tissue injuries. However, it is unlikely that retrospective case–control genetic association studies can define the variant landscape of such multifactorial conditions.

Prediction of injury predisposition based on DNA testing is likely to be of limited value at present, and the higher-order ‘bioassay’ tests are likely to remain a key element of predisposition identification for the foreseeable future [133]. The use of DNA testing is unlikely to ever surpass the predictive capacity of a field test, but it can be used in conjunction with field and laboratory testing to increase the likelihood of identifying athletes who are more prone to injury.

Detailed results obtained from whole exome and genome sequencing (WES/WGS) of patients with inherited diseases have challenged the classical definition of genetic causation and the concept of strictly monogenic disorders [134,135,136]. It is presumed that oligo- and polygenic inheritance, where the causative mutation is accompanied by multiple phenotype-modifying or co-causative variants, may account for a significant proportion, especially atypical cases, of hereditary diseases and normal phenotypes. The concept of polygenic inheritance not only changes the methodology for identifying causal gene mutations, but also affects genetic diagnosis and genetic counseling [137]. Therefore, distinguishing between possible co-factors or modifying variants and the thousands of irrelevant variants identified in WES/WGS has become one of the biggest challenges facing medical and sport genetics. This requires the use of advanced bioinformatics methods (e.g., machine learning-based prediction algorithms) and basic research that identifies the functional effects of gene variants (or, more precisely: the products of genes carrying said variants) at multiple levels of organization of biological systems simultaneously (multiomics) to identify the key biological pathways contributing to injury susceptibility, and bring us closer to a better understanding of the genetic mechanisms underlying predisposition to injury.

It is certainly too early to determine the clinical implications of genetic variability as a tool for improving current methods of training and injury prevention. Therefore, a multidisciplinary approach should be adopted, utilizing the power of next-generation technologies in conjunction with functional expression studies, animal models, and bioinformatics analysis tools to further characterize the genetic risk profile of musculoskeletal soft tissue injuries [7]. As recently demonstrated by Collins and September (2023), the currently available evidence does not support the marketing of a commercial genetic test to determine susceptibility to musculoskeletal injuries. Our review indicates that basic research into the association of genomic variants with sport-related musculoskeletal pathology can and should be conducted. However, the results to date have very limited relevance to the diagnosis of specific athletes, even in a well-established clinical setting [138].

## Figures and Tables

**Table 1 cells-13-01828-t001:** The most commonly analysed gene variants associated with exercise-induced muscle damage.

Gene/Variant	Beneficial	Participants	Ethnicity	Discipline	Sex		Ref.	Contents/Assessment
*ACTN3* (α-actinin skeletal muscle isoform 3) rs1815739(C/T) (R577X)	CC	64	Italian	top football players	-	M	[87] *	injuries recorded in 10 seasons
	-	100	Spanish Caucasians	elite endurance athletes	F	M	[36] $	injuries recorded in one season
	CC	83	Brazilian	professional soccer players	-	M	[48]	injuries recorded in 3 seasons
	-	46	Australian	elite football players	-	M	[10] !	injuries recorded in 7 seasons
	-	122	Spanish	professional soccer players	-	M	[92] %	injuries recorded in one season
	CC	402	Caucasians	academy soccer players	-	M	[1] @	severity and type of injuries recorded in 3 seasons
	CC	46	Spanish	top-level professional soccer players	F	M	[30]	injuries and recovery time recorded in 5 seasons
	CC	22	Spanish Caucasians	experienced triathletes	F	M	[93] ^	low vs. high CK response after the half-ironman
	-	139	Spanish	amateur marathoners	F	M	[32]	injuries recorded in one season
	CC	8	Caucasians	4RR and 4XX individuals	-	M	[94]	biopsies and blood analysis after intensive eccentric knee flexion
	CC	43	Spanish	professional football players	-	M	[29]	injuries and recovery time recorded in 7 seasons
	TT	30	Brazilian	young soccer players (under 16y)	-	M	[95]	CK and hormone levels after the game
	CC	169	Italian	professional football players	-	M	[96]	injuries recorded in 5 seasons
	CC	76	Japan	physical education students	-	M	[97]	increased passive muscle stiffness of the hamstring
	CC	20	Brazilian	ultra-runners	F	M	[98]	CK among other proteins after the adventure race
	-	67	Spanish Caucasians	marathon runners	F	M	[19] #	low vs. high CK response after the marathon race
	CC	23	Spanish Caucasians	experienced triathletes	F	M	[27] ^	CK level and leg muscle power reduction after the half-ironman race
	CC	71	Spanish Caucasians	volunteer marathon runners	F	M	[99]	CK level and leg muscle power reduction after the marathon
	TT	99	Japan	physical education students	F	-	[100]	According to the injury history reports
	CC	442	Chinese Han	the infantry division	-	M	[101]	142 patients with ankle sprains
	CC	300	Korea	97 elite ballerinas and 203 controls	F	-	[102]	ankle injury and low flexibility
	CC	37	Brazilian	soccer players	-	M	[26]	CK among other proteins, hormones, inflammatory responses after the eccentric training
	CC	452	Chinese	military	-	M	[103]	handgrip strength
	TT	17	Lithuanian Caucasians	active volunteers	-	M	[104]	the response of muscle function to plyometric jumping exercise
	CC	181	128 Caucasians, 32 African-Americans, 9 Hispanic, 12 Asian	47 ER cases, 134 active volunteers, and military	F	M	[20] ~	CK level after a standardized exercise challenge (included stepping up and down with load)
	CC	19	Lithuanian Caucasians	active volunteers	-	M	[24]	muscle biopsies and blood analysis after maximal eccentric knee extensions
	-	156	115 Caucasians, 4 African-Americans, 6 Hispanic, 20 Asian, 11 other	volunteers	F	M	[105]	CK level and maximal voluntary contraction after elbow flexion eccentric exercise protocol
	CC	602	469 Caucasians, 28 African-Americans, 25 Hispanic, 55 Asian	volunteers	F	M	[106]	muscle size and strength after standardized elbow flexor/ extensor resistance training program
*CK-MM* (muscle-specific creatine kinase) rs8111989 (A/G) (NcoI)	-	100	Spanish Caucasians	elite endurance athletes	F	M	[36] $	injuries recorded in one season
	-	122	Spanish	professional soccer players	-	M	[92] %	injuries recorded in one season
	-	22	Spanish Caucasians	experienced triathletes	F	M	[93] ^	low vs. high CK response after the half-ironman
	-	67	Spanish Caucasians	marathon runners	F	M	[19] #	low vs. high CK response after the marathon race
	AA	181	128 Caucasians, 32 African-Americans, 9 Hispanic, 12 Asian	47 ER cases, 134 active volunteers, and military	F	M	[20] ~	CK level after a standardized exercise challenge (included stepping up and down with load)
	GG	88	76 Caucasians, 12 non-Caucasians	physically active volunteers	F	M	[37]	CK level after a standardized exercise challenge (included stepping up and down with a load
*AMPD1* (C34T) (adenosine monophosphate deaminase 1) rs17602729(C/T)	CC	100	Spanish Caucasians	elite endurance athletes	F	M	[36] $	injuries recorded in one season
	CC	122	Spanish	professional soccer players	-	M	[92] %	injuries recorded in one season
	-	181	128 Caucasians, 32 African-Americans, 9 Hispanic, 12 Asian	47 ER cases, 134 active volunteers, and military	F	M	[20] ~	CK level after a standardized exercise challenge (included stepping up and down with load)
*MLCKII* (myosin light chain kinase) rs28497577(C/A) (C37885A; P21H)	-	100	Spanish Caucasians	elite endurance athletes	F	M	[36] $	injuries recorded in one season
	AA	122	Spanish	professional soccer players	-	M	[92] %	injuries recorded in one season
	CC	402	Caucasians	academy soccer players	-	M	[1] @	severity and type of injuries recorded in 3 seasons
	-	22	Spanish Caucasians	experienced triathletes	F	M	[93] ^	low vs. high CK response after the half-ironman
	-	67	Spanish Caucasians	marathon runners	F	M	[19,41] #	low vs. high CK response after the marathon race
	CC	181	128 Caucasians, 32 African-Americans, 9 Hispanic, 12 Asian	47 ER cases, 134 active volunteers, and military	F	M	[20] ~	CK level after a standardized exercise challenge (included stepping up and down with load)
	CC	156	115 Caucasians, 4 African-Americans, 6 Hispanic, 20 Asian, 11 other	volunteers	F	M	[105]	CK level and maximal voluntary contraction after elbow flexion eccentric exercise protocol
*MLCKI* (myosin light chain kinase) rs2700352(C/T) (C49T)	-	100	Spanish Caucasians	elite endurance athletes	F	M	[36] $	injuries recorded in one season
	-	122	Spanish	professional soccer players	-	M	[92] %	injuries recorded in one season
	-	181	128 Caucasians, 32 African-Americans, 9 Hispanic, 12 Asian	47 ER cases, 134 active volunteers, and military	F	M	[20] ~	CK level after a standardized exercise challenge (included stepping up and down with load)
	CC	156	115 Caucasians, 4 African-Americans, 6 Hispanic, 20 Asian, 11 other	volunteers	F	M	[105]	CK level and maximal voluntary contraction after elbow flexion eccentric exercise protocol
*ACE* (angiotensin I converting enzyme) rs4341 (I/D)	DD	64	Italian	top football players	-	M	[87] *	injuries recorded in 10 seasons
	-	100	Spanish Caucasians	elite endurance athletes	F	M	[36] $	injuries recorded in one season
	DD	83	Brazilian	professional soccer players	-	M	[48]	injuries recorded in 3 seasons
	-	122	Spanish	professional soccer players		M	[92] %	injuries recorded in one season
	- DD	341 369	Italian Japanese	football players	- -	M M	[44]	injuries recorded in 10 seasons
	-	22	Spanish Caucasians	experienced triathletes	F	M	[93] ^	low vs. high CK response after the half-ironman
	DD	81	Brazilian	volunteers endurance runners	-	M	[45]	CK and inflammatory markers after the marathon
	-	67	Spanish Caucasians	marathon runners	F	M	[19] #	low vs. high CK response after the marathon race
	-	300	Korea	97 elite ballerinas, 203 controls	F	-	[102]	ankle injury/low flexibility
	-	181	128 Caucasians, 32 African-Americans, 9 Hispanic, 12 Asian	47 ER cases, 134 active volunteers, and military	F	M	[20]~	CK level after a standardized exercise challenge (included stepping up and down with load)
	DD	70	Spanish Caucasians	physical education students	F	M	[47]	CK level after eccentric contractions of the elbow flexor muscles
	-	88	76 Caucasians,12 non-Caucasians	physically active volunteers	F	M	[37]	CK level was measured after an exercise challenge (included stepping up and down with load)
*CCL2* (chemokine ligand 2) rs2857656 (G/C)	-	46	Australian	elite football players	-	M	[10] !	injuries recorded in 7 seasons
	-	402	Caucasians	academy soccer players	-	M	[1] @	severity and type of injuries recorded in 3 seasons
	CC	73	43 White, 11 Black Africans, 19 Hispanics	professional soccer players	-	M	[52]	injuries recorded in 3 seasons
*IL6* (interleukin-6) rs1800795 (-174G>C)	-	466	South African Caucasian	232 controls, 234 ACLR (including 135 non-contact)	F	M	[107]	Injuries recorded in 8 seasons
	GG	402	Caucasians	academy soccer players	-	M	[1] @	severity and type of injuries recorded in 3 seasons
	-	22	Spanish Caucasians	experienced triathletes	F	M	[93] ^	low vs. high CK response after the half-ironman
	CC	423	Polish Caucasians	229 ACLR soccer players, 194 active controls	F	M	[108]	Injuries recorded in 8 seasons
	CC	107	Spanish	elite soccer players	-	M	[57] &	hamstring injuries recorded in 6 seasons
	-	67	Spanish Caucasians	marathon runners	F	M	[19] #	low vs. high CK response after the marathon race
	-	466	South African Caucasian	232 active controls, 234 ACLR (including 135 non-contact)	F	M	[109]	injuries recorded in 8 seasons
	-	544	251 Caucasian South African, 293 Caucasian Australian	175 AT, 369 physically active controls	F	M	[110]	
	-	181	128 Caucasian, 32 African-Americans, 9 Hispanic, 12 Asian	47 ER cases, 134 physically active volunteers, and military	F	M	[20] ~	CK level after a standardized exercise challenge (included stepping up and down with load)
	GG	70	Israeli Caucasians	physical education students	F	M	[58]	CK level after elbow flexor eccentric exercise protocol
TNFα (tumor necrosis factor) rs1800629(G/A) (-308G>A)	-	22	Spanish Caucasians	experienced triathletes	F	M	[93] ^	low vs. high CK response after the half-ironman
	-	67	Spanish Caucasians	marathon runners	F	M	[19] #	low vs. high CK response after the marathon race
		466	South African Caucasian	232 active controls, 234 ACLR (including 135 non-contact)	F	M	[109]	injuries recorded in 8 seasons
	-	70	Israeli Caucasians	physical education students	F	M	[58]	CK level after elbow flexor eccentric exercise protocol
*IGF2* (insulin-like growth factor II) rs3213221(C/G) (C13790G)	GG	46	Australian	elite football players	-	M	[10] !	Injuries recorded in 7 seasons
	-	22	Spanish Caucasians	experienced triathletes	F	M	[93] ^	low vs. high CK response after the half-ironman
	-	67	Spanish Caucasians	marathon runners	F	M	[19] #	low vs. high CK response after the marathon race
	CG	73	43 White, 11 Black Africans, 19 Hispanics	professional soccer players		M	[52]	injuries recorded in 3 seasons
	GG	156	115 Caucasians, 4 African-Americans, 6 Hispanic, 20 Asian, 11 other	Volunteers	F	M	[111]	CK level and maximal voluntary contraction after elbow flexion eccentric exercise protocol
*IGF2* (insulin-like growth factor II) rs680(G/A) (G17200A; ApaI)	GG	156	115 Caucasians, 4 African-Americans, 6 Hispanic, 20 Asian, 11 other	Volunteers	F	M	[111]	CK level and maximal voluntary contraction after elbow flexion eccentric exercise protocol
	GG	579	497 Caucasians, 70 African-Americans, 12 other	volunteers of different ages	F	M	[112]	arm and leg strength and body composition
	GG	693	Caucasian-UK	older volunteers	F	M	[113]	grip strength
*MMP3* (Matrix metallo -proteinase 3) rs679620 (C/T)	CC	402	Caucasians	academy soccer players	-	M	[1] @	severity and type of injuries recorded in 3 seasons
	TT	421	Polish Caucasians	229 ACLR soccer players, 192 active controls	F	M	[70]	injuries recorded in 8 seasons
	CC	107	Spanish	elite soccer players	-	M	[57] &	hamstring injuries recorded in 6 seasons
	CC	274 466	Australian South African	79 AT, 195 controls 234 ACLR, 232 controls	F	M	[71]	a significant risk for AT
	-	74	Spanish	elite soccer players		M	[114]	muscle injuries recorded in 5 seasons
	TT	249	British Caucasians	Clinic patients: 93 AT, and 25 ATR; 131 active controls	F	M	[69]	a significant risk for ATR
	CC	345	South African Caucasians	129 ACLR, and 216 active controls	F	M	[72]	
	TT	212	South African Caucasians	75 AT, 39 ATR, and 98 controls			[68]	a significant risk for AT
*TNC* (Tenascin C) rs2104772 (A/T)	-	421	Polish Caucasian	229 ACLR soccer players, 192 active controls	F	M	[115]	injuries recorded in 8 seasons
	TT	107	Spanish	elite soccer players	-	M	[57] &	hamstring injuries recorded in 6 seasons
	TT	20	South African	10 AT	-	M	[116]	exome sequencing and a customized pipeline
	-	302	Caucasians	patients after mini-open rotator cuff repair	F	M	[78]	the integrity of the repair 1 year postoperatively with an ultrasound linear array transducer
	TT	125 293	South African Caucasians Australian Caucasians	94 AT; 31 active controls 85 AT; 208 active controls	F	M	[76]	
*COL5A1* (Collagen Type V Alpha 1 Chain) rs12722 (C/T)	-	64	Italian	top football players	-	M	[87] *	injuries recorded in 10 seasons
	CC	46	Australian	Elite football players	-	M	[10] !	injuries recorded in 7 seasons
	TT	402	Caucasians	academy soccer players	-	M	[1] @	severity and type of injuries recorded in 3 seasons
	CC	347	Turkey	154 patients with tennis elbow, 195 active controls	F	M	[86]	patients performing repetitive activities or heavy work
	-	54	Italian	professional soccer players	-	M	[117]	injuries recorded in 4 seasons
	-	321	Polish Caucasians	recreational skiers: 138 ACLR; 183 controls	-	M	[118]	
	-	300	Korea	97 elite ballerinas; 203 controls	F	-	[102]	an ankle injury/low flexibility
	CC	73	43 White, 11 Black Africans, 19 Hispanics	professional soccer players	_	M	[52]	injuries recorded in 3 seasons
	CC	345	Caucasian South Africans	129 ACLR, 216 active controls	F	M	[89]	
	CC	212	South African Caucasians	75 AT, 39 ATR, 98 controls			[68]	a significant risk for AT
	CC	520	295 Australian; 225 Caucasian South Africans	178 AT; 342 Controls	F	M	[85]	physically active patients recruited from the clinics
	CC	240	South African Caucasians	111 AT; 129 active controls	F	M	[84]	physically active patients recruited from the clinics
*COL1A1* (Collagen Type I Alpha 1 Chain) rs1800012 (G/T)	GG	46	Australian	elite football players	-	M	[10] !	injuries recorded in 7 seasons
	-	402	Caucasians	academy soccer players	-	M	[1] @	severity and type of injuries recorded in 3 seasons
	_	251	South African Caucasians	85 AT, 41 ATR, 125 controls	F	M	[119]	patients recruited from the clinics
	-	206	Turkey	103 patients with tennis elbow, 103 controls	F	M	[120]	patients recruited from the clinics
	_	73	43 White, 11 Black Africans, 19 Hispanics	professional soccer players	_	M	[52]	injuries recorded in 3 seasons
	-	234	Polish Caucasians	professional soccer players (91 with ACLR)	-	M	[90]	
	GG	321	Polish Caucasians	recreational skiers: 138 ACLR; 183 controls	F	M	[91]	
	TT	247	South African Caucasian	117 ACLR (direct contact in 15 patients), 130 active controls	F	M	[121]	physically active patients recruited from the clinics
	TT	684	Sweden	233 ACLR, 126 shoulder dislocation; 325 controls	F	M	[88]	patients recruited from the clinics

CK—creatine kinase; ER—exertional rhabdomyolysis, AT—Achilles tendinopathy; ATR—Achilles tendon rupture; ACLR—anterior cruciate ligament rupture. Multifactor analysis is included in the studies: * (Massidda et al., 2024 [87]) *ACE* I/D, *ACTN3* rs1815739, *COL5A1* rs2722, *MCT1* rs1049434; $ (Varillas-Delgado et al., 2023 [36]) *ACE* I/D, *ACTN3* rs1815739, *CKM* rs8111989, *AMPD1* rs17602729, *MLCK* rs2849757, *MLCK* rs2700352; % (Maestro et al., 2022 [92]) *ACE* I/D, *ACTN3* rs1815739, *CKM* rs8111989, *AMPD1* rs17602729, *MLCK* rs28497577, *MLCK* rs2700352; ! (Jacob et al., 2022 [10]) *ACTN3* rs1815739, *IGF2* rs3213221, *CCL2* rs2857656, *COL1A1* rs1800012, *COL5A1* rs12722, *COL12A1* rs970547, *EMILIN1* rs2289360, *NOGGIN* rs1372857, *SMAD6* rs2053423; @ (Hall et al., 2021 [1]) *ACTN3* rs1815739, *MLCK* rs28497577, *IL6* rs1800795, *COL5A1* rs12722, *COL1A1* rs1800012, *CCL2* rs2857656, *EMILIN1* rs2289360, *MMP3* rs28497577, *VEGFA* rs2010963; ^ (Del Coso et al., 2020 [93]) *ACE* I/D, *ACTN3* rs1815739, *CKMM* rs1803285, *MLCK* rs28497577, *TNFα* rs1800629, *IGF2* rs3213221, *IL6* rs1800795; & (Larruskain et al., 2018 [57]) *MMP3* rs679620, *TNC* rs2104772, *IL6* rs1800795; *NOS3* rs1799983, *HIF1A* rs11549465; # (Del Coso et al., 2017 [19]) *ACE* I/D, *ACTN3* rs1815739, *CKMM* rs1803285, *MLCK* rs28497577, *TNFα* rs1800629,*IGF2* rs3213221, *IL6* rs1800795; ~ (Deuster et al., 2013 [20]) *ACE* I/D, *ACTN3* rs1815739, *CKMM* rs1803285, *MLCK* rs28497577, *MLCK* rs2700352, *AMPD1* rs17602729, *IL6* rs1800795, *HSPA1B* rs1061581.

## Data Availability

Not applicable.

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
