# Peer review of "The Genetic Basis of Non-Contact Soft Tissue Injuries-Are There Practical Applications of Genetic Knowledge?"

_cells, 2024, doi:10.3390/cells13221828_

Round 1

Reviewer 1 Report

Comments and Suggestions for Authors

The manuscript is comprehensive and updated.

The main question addressed by the research is the usefulness of genetic testing to evaluate injury (and/or recovery) predisposition of athletes related to musculoskeletal structures (muscles, tendons, and ligaments).  The conclusion of the review is that all published studies so far are contradictory in results, both when they were carried out on single genes and on multiple variants (using polygenic risk score or total genotype score). As such, at present, the genetic tests have no clinical value or utility and contradict the assertion of several companies that sell on the web direct-to-consumer (DTC) genetic testing to predict sports performance/injury risk associated with sport.

In my opinion, the proposed review is comprehensive, original and is useful in the field of genetic testing applied to athletic performance, confirming that the marketing of commercial genetic tests to determine susceptibility to musculoskeletal injuries is meaningless.

The manuscript offers an updated overview of the subject, underlying also the limits of finding thousands of irrelevant variants using novel technologies, such as whole genome sequencing (WGS) and whole exome sequencing (WES).

In the conclusion the authors mention also the support that machine learning-based prediction algorithms could give to the field. Maybe the authors could stress more in detail how the AI could help in identifying clinically relevant variants in different genes after a WGS analysis.

The conclusions are consistent with the evidence reported so far in literature. The references are appropriate. No additional comments on the table.

Author Response

Thank you very much for taking the time to review this manuscript. We sincerely appreciate the
Reviewer's insightful comments and valuable suggestions for improving our manuscript. We are
grateful for the positive feedback regarding our work.

Please find the detailed response below:

Comments 1: In the conclusion the authors mention also the support that machine learningbased
prediction algorithms could give to the field. Maybe the authors could stress more in
detail how the AI could help in identifying clinically relevant variants in different genes after a
WGS analysis.

The role of AI is becoming increasingly important in prioritizing variants based on methods and
techniques such as pathogenicity prediction (e.g. SIFT, PolyPhen), conservation scores, effect on
protein structure and function, phenotype-guided filtering, gene-disease association networks,
population frequency and family studies, significantly enhancing the efforts of bioinformatics
teams. The importance of AI in integrating multi-omics data to provide a comprehensive
understanding of how variants may affect biological pathways and contribute to disease cannot
be overstated, helping to identify patterns and assess variant significance, and providing realtime
support and recommendations based on the latest research and clinical guidelines.
Although the Reviewer's suggestion seems very valuable, given that the extensive potential of AI
in genetic variant analysis, we have chosen not to explore this topic in depth in our manuscript,
which focuses on the current state of knowledge. Furthermore, none of the authors consider
themselves experts in the vast field of bioinformatics.

Reviewer 2 Report

Comments and Suggestions for Authors

This manuscript reviews the most commonly researched genetic variants that are thought to predispose athletes to soft tissue injuries.

The authors concluded that it is too early to determine the clinical implications of genetic variability as a tool for improving well-established training and injury prevention methods, as the predictive power of genetic testing for injury predisposition is currently low.

I appreciate the authors reviewing this important issue in the sports medicine field. This review may become a fundamental paper for genetics in sports medicine.

Author Response

Thank you very much for taking the time to review this manuscript. We are very
grateful for the positive feedback regarding our work.

Reviewer 3 Report

Comments and Suggestions for Authors

After reviewing the article entitled "The Genetic Basis of Non-Contact Soft Tissue Injuries: Are There Practical Applications of Genetic Knowledge?" by Borzemska et al. submitted for possible publication in Cells, I would like to offer the following comments:

1. The sequential structure of the review is well thought out and each topic is thoroughly detailed.

2. The authors provide a comprehensive and well-reasoned review of the current and potential use of genetic tools in the diagnosis of musculoskeletal injuries in athletes. They aptly conclude that it is premature to implement these tools in real-world sports medicine, while also highlighting new avenues of research needed to improve our understanding of genetic markers associated with musculoskeletal injuries in athletes.

3. I respectfully suggest that the title be changed to "The Genetic Basis of Non-Contact Soft Tissue Injuries: Practical Applications of Genetic Findings" to improve clarity and focus.

4. In addition, the inclusion of figures in the manuscript would enhance reader comprehension and engagement.

Overall, the review is well structured and insightful, and these suggestions may further strengthen the manuscript.

Author Response

Thank you very much for taking the time to review this manuscript. We sincerely appreciate the
Reviewer's insightful comments and valuable suggestions for improving our manuscript. We are
grateful for the positive feedback regarding our work.

Please find the detailed responses below:

Comments 1: I respectfully suggest that the title be changed to "The Genetic Basis of Non-
Contact Soft Tissue Injuries: Practical Applications of Genetic Findings" to improve clarity and
focus.

Thanks to the Reviewer for pointing this out. However, we have decided to retain the title in its
current form as a question. We believe that this phrasing effectively reflects the uncertainties
surrounding the application of genetic knowledge in assessing predisposition to soft tissue
injuries. This approach may help to set appropriate expectations for the reader, indicating that
the article explores these complexities rather than providing definitive guidelines for practical
application.

Comments 2: In addition, the inclusion of figures in the manuscript would enhance reader
comprehension and engagement.

We appreciate the Reviewer’s insights on improving readability. After careful consideration, we
have decided not to include additional figures, as we aim to maintain a concise presentation that
focuses on the core results. Detailed data is included in the table. We believe that the current
format adequately conveys our findings and supports our conclusions.